# An Increase in Fat-Free Mass is Associated with Higher Appetite and Energy Intake in Older Adults: A Randomised Control Trial

**DOI:** 10.3390/nu13010141

**Published:** 2021-01-01

**Authors:** Kelsie Olivia Johnson, Adrian Holliday, Nathan Mistry, Andrew Cunniffe, Kieran Howard, Nicholas Stanger, Lauren L. O’Mahoney, Jamie Matu, Theocharis Ispoglou

**Affiliations:** 1Carnegie School of Sport, Leeds Beckett University, Leeds LS6 3QS, UK; Adrian.holliday@newcastle.ac.uk (A.H.); n.mistry6@herts.ac.uk (N.M.); andycunniffe@live.com (A.C.); kieran.howard@live.com (K.H.); N.Stanger@leedsbeckett.ac.uk (N.S.); llom1@leicester.ac.uk (L.L.O.); J.Matu@leedsbeckett.ac.uk (J.M.); T.Ispoglou@leedsbeckett.ac.uk (T.I.); 2Higher Education Sport, Hartpury University, Hartpury GL19 3BE, UK; 3Human Nutrition Research Centre, Population Health Sciences Institute, Newcastle University, Newcastle NE1 7RU, UK; 4School of Life and Medical Sciences, University of Hertfordshire, Hatfield AL10 9AB, UK; 5Diabetes Research Centre, Leicester General Hospital, University of Leicester, Leicester LE5 4PW, UK; 6School of Clinical & Applied Sciences, Leeds Beckett University, Leeds LS1 3HE, UK

**Keywords:** fat-free mass, ageing, older adults, appetite, energy intake, protein intake, resistance training

## Abstract

Cross-sectional studies in younger adults have demonstrated a positive association between energy intake (EI) and fat-free mass (FFM), with this relationship seemingly mediated by resting metabolic rate (RMR). Establishing a causal effect longitudinally would be prudent in older adults suffering from loss of appetite. We investigated the effects of FFM on RMR, appetite and EI in 39 healthy older adults (age: 66 ± 4 years, BMI: 25.1 ± 3.5 kg∙m^2^) assigned to either 12-week resistance training + protein supplementation group (RT + PRO) or control group (CON). Body composition, subjective appetite, leptin, insulin, RMR and laboratory-measured *ad libitum* EI were measured at baseline, weeks 6 and 12 of the intervention, while daily EI at baseline and week 12. FFM (+1.2 kg; *p* = 0.002), postprandial subjective appetite (+8 mm; *p* = 0.027), *ad libitum* EI (+119 kcal; *p* = 0.012) and daily EI (+133 kcal; *p* = 0.010) increased from baseline to week 12 in the RT + PRO. RMR, fasted subjective appetite, leptin and insulin concentrations remained unchanged (all *p* > 0.05). The increases *ad libitum* EI correlated with increases in FFM (r = 0.527, *p* = 0.001), with 54% of the change in EI attributed to FFM changes. In conclusion, FFM increases were associated with an increased *ad libitum* EI and postprandial appetite in older adults.

## 1. Introduction

Older adults typically experience reductions in energy intake (EI) as a result of an age-associated reduction in appetite—a phenomenon termed the “anorexia of ageing” [1]. The anorexia of ageing is the driver of unintentional weight loss, particularly to skeletal muscle tissue through protein malnutrition [2,3]. Energy intake declines at a greater rate than energy expenditure causing an imbalance in homeostatic appetite control and thus, body weight [4]. The appetite-regulatory system is complex and multifaceted [5]. We have recently conducted a meta-analysis which identified that concentrations of the anorexigenic hormone leptin, insulin, CCK, and PYY (postprandial only) were higher in older adults in comparison to younger adults [6]. This may partially explain reductions in hunger perceptions and EI in older adults.

Resting metabolic rate contributes ~60–70% of total energy expenditure and decreases by ~1–2% per decade in adults [7,8,9,10], which is aligned to the 1–2% reduction in muscle mass observed with ageing. In addition, resting metabolic rate has been strongly correlated with self-determined meal size, self-reported meal frequency and daily hunger across a 12-week physical activity intervention in overweight/obese individuals [11,12]. Fat-free mass (FFM) is the greatest contributor to resting metabolic rate accounting for 60–70% [7] and, research has demonstrated positive associations with self-selected meal size, *ad libitum* and total daily EI [13,14,15,16]. Indeed, path analysis has demonstrated that such associations between FFM and EI is mediated by resting metabolic rate; FFM and RMR together attributed to 62% of the variance in EI [17]. This suggests that FFM is the main determinant of resting metabolic rate and that metabolic requirements drive EI. It is proposed that the drive to eat may be underpinned by a protective mechanism to ensure that EI is not below energetic requirements [18,19] and thus, it is likely that any increase in FFM would increase EI through the known associations between resting metabolic rate and EI shown by previous research [11,13,14]. However, these studies are correlational and observational thus, cause and effect cannot be established. Therefore, manipulating FFM to determine consequent changes in RMR, appetite and EI would be of interest, particularly for older adults given that both reduced appetite and FFM are observed with ageing.

Physical activity/exercise can raise energy expenditure several fold depending on the volume, mode and intensity; however, physical activity is not a potent nor stable driver of EI [20]. Evidence suggests that exercise-induced alterations in energy expenditure do not significantly influence EI in the immediate-post exercise period [21]. Additionally, any suppression in appetite through the phenomenon of exercise-induced anorexia is usually alleviated within 30 min after exercise cessation [22,23]. The same was demonstrated in unpublished data in our laboratory for older adults following resistance exercise (RE), whereby appetite and EI were unaffected in the two-hour post-exercise period compared to the non-exercising control group. However, exercise conducted over an extended period from 9–16 days have unveiled partial compensatory responses [24,25]. These changes have also been observed in appetite-related hormone concentrations under conditions of weight loss [26]. The compensatory responses are consistent with the proposition that energy expenditure acts as a driver for EI. However, the association between exercise and EI is weaker than the relationship of resting metabolic rate with EI, likely due to resting metabolic rate providing a greater contribution to total energy expenditure [19].

Currently, no research to date has manipulated body composition, in particular, FFM experimentally to determine the effects on resting metabolic rate and thus appetite and EI longitudinally. Therefore, the purpose of this study was to promote gains in muscle mass—through the established approach of resistance training and protein supplementation [27,28]—to manipulate FFM and determine the longitudinal responses in RMR, appetite and EI in older adults prone to reductions in muscle mass and anorexia of ageing.

## 2. Materials and Methods

### 2.1. Participants

Forty-six participants were initially enrolled on to the study after initial screening. Following withdrawals (see Figure 1) 39 healthy, independently living older adults (22F, 17M; Table 1) were included in the study. Participants were recruited if they met the following inclusion criteria: Aged 60 years or over, do not smoke, no history of cardiovascular disease, metabolic disease or dyslipidaemia, not dieting within the last 3 months, not taking drugs known to affect digestion or metabolism, participated in at least “some activity” according to physical activity classification domains, weight stable for the last 3 months, i.e., <2.3 kg change in body weight [29], tolerance and no allergies for food items and ingredients provided during the experimental trials and not currently following a structured resistance training programme or have recently participated in structured resistance exercise within the past 12 months.

The study obtained approval from the Leeds Beckett University Research Ethics Committee and was conducted in accordance with the Declaration of Helsinki (Application reference: 58906).

### 2.2. Experimental Design

A 15-week randomised control trial was conducted with the inclusion of two experimental groups (Figure 2). Two weeks of pre-testing (baseline) were conducted prior to the 12-week intervention which involved random allocation to either a control group (CON) or resistance training and protein supplementation group (RT + PRO). The immediate week following the completion of the 12-week intervention involved the post-testing. Participants were required to attend a total of six laboratory visits. Participants recorded their food intake for the 24 h prior to the first experimental trial; the quantity and timing of this intake were then replicated before each subsequent trial. Participants in both groups arrived at the laboratory after an overnight fast of ≥10 h and were required to avoid alcohol, caffeine and strenuous exercise in the 24 h preceding the trial. Two hours before the laboratory testing participants were required to consume 300 mL of water to ensure euhydration. Additional visits to the University gym were required for the participants who were assigned to the RT + PRO.

### 2.3. Pre-Testing (Baseline)

On the first week of baseline testing, anthropometric variables were recorded. Due to the difficulty in the measurement of basal metabolic rate in a laboratory setting, resting metabolic rate was assessed instead. Previous research has shown that resting metabolic rate is typically 3% higher than basal metabolic rate [30,31]. Body composition assessment was subsequently conducted. Following this, all participants were familiarised to the standardised test meals and resistance exercise machines included in the randomised control trial. Participants were given verbal and written instructions on how to complete a three-day food diary for the assessment of daily EI. Participants were then provided with an example and a blank template document to complete the weighed food diary.

After the body composition measurement participants were matched for FFM scores and allocated into CON or RT + PRO. Block randomisation was used to assign equal sample numbers to each group of four participants (block). The block size was in fours which created a possible six sequences: these were AABB, ABAB, ABBA, BAAB, BABA, BBAA. This was repeated until the 46 participants were all assigned. This was done to ensure that baseline FFM would not be a confounding variable [32,33]. In the second visit of baseline testing, following the measurements of fasted subjective appetite and appetite hormone concentrations, a meal tolerance test (MTT) was conducted to assess postprandial appetite-related hormone concentrations, subjective appetite and *ad libitum* EI. Participants then completed a five-repetition maximum (5-RM) test to assess maximum strength. The 5RM scores were used to estimate the required workloads during the resistance exercise sessions for those assigned to RT + PRO. These initial preliminary visits were separated by ≥7 days to prevent any influence of unaccustomed exercise on appetite and EI.

### 2.4. Mid and Post-Testing (Week 6 and Week 12)

All procedures conducted during pre-testing were repeated, in the same sequence, at week 6 and within 4 days of the final resistance training session (week 12), with the exception of the 5-RM and daily EI measures, which were repeated only at week 12. Familiarisation sessions were not required at weeks 6 and 12. The procedures in week 12 were identical to week 6; however, a 5-RM was conducted following the MTT (see Figure 2). In all instances, resting metabolic rate was not conducted within the 48 h window following resistance exercise.

### 2.5. Experimental Protocol

Participants assigned to RT + PRO were required to visit Leeds Beckett University gym twice per week, with at least 48 h between training sessions, for a duration of 12 weeks. Participants were required to complete resistance training sessions twice weekly in accordance with the current ACSM guidelines [34]. The order in which the seven exercises were performed was seated row, leg press, chest press, leg extension, lat pull down, leg curl and shoulder press. For the first two weeks of the intervention, participants conducted 2 sets of 10–15 reps at 65–75% of the predicted 1-RM. Due to safety reasons 5-RM was to predict 1-RM [35]. This intensity was selected as research has shown intensities ranging from 65–75% result in enhancements in skeletal muscle mass in older adults [34]. From week three of the intervention, the number of sets performed for each exercise increased from two to three with a repetition range of 10–15. Two minutes’ rest was provided between each set. For each of the aforementioned exercises, 5-RM strength was reassessed at week 4 and week 8 at the end of the four-week training cycle, and subsequently, the training load was adjusted accordingly. Each four-week training cycle started at 65% of the new predicted 1-RM and the training load was increased to promote progressive overload of skeletal muscle. All sessions were supervised by a qualified personal trainer.

Additionally, participants received two portions of whey protein (USN Blue Lab Whey) per day, which contained 15 g of protein each (30 g total, 169 kcal per day). This portion was selected as previous research has shown two 15 g portions of protein was enough to stimulate a 1.3 kg increase in lean body mass in older adults alongside 12 weeks of resistance training [28]. Estimates of meal-level dietary protein intake find lower protein-containing meals at breakfast and lunch which are suboptimal concerning stimulation of MPS, with only one higher protein-containing meal (normally dinner, 30–40 g protein) reaching a protein intake that is optimal for postprandial MPS [36,37]. Therefore, participants consumed 15 g of protein with breakfast and lunch. Participants assigned to CON were instructed to continue their current lifestyle as usual with no changes to physical activity or protein/EI.

### 2.6. Outcome Measures

#### 2.6.1. Resting Metabolic Rate

Resting metabolic rate was measured via indirect calorimetry using a ventilated hood connected to a canopy (Cortex, Canopy, Leipzig, Germany) connected to an online gas analysis system (Metalyser 3B, Leipzig, Germany). Twenty-four hours before visiting the laboratory participants were told to avoid exercise. On the morning of experimental trials, participants used motorised transport where possible. Participants also used the lift and avoided the stairs to the laboratory to minimise exertion. Prior to use, the Metalyser was calibrated using reference gases, daily barometric pressure, and a 3 L volume syringe. Calibration gas of a known oxygen (15%) and carbon dioxide (5%) content was used, with calibration accepted as a deviation of <±0.1%. Testing was conducted in accordance with guidelines established by Compher and colleagues [38]. Before the assessment of resting metabolic rate, participants were rested for 10 min in a dim lit room. After the rest period, the ventilated hood, which was connected to the Metalyser, was placed over the head of the participant, with the surrounding ‘skirt’ secured in place. During the measurement period, air flowed continuously into the hood, over the participant’s head, whilst expired air was extracted at the same rate into the Metalyser. Participants remained quiet and still, but awake, in the supine position through the entire resting metabolic rate procedure. The room was maintained in low light, and noise was kept at a minimum [39,40]. The average room temperature across trials was 23.3 ± 1.6 °C. The expired fraction of oxygen and carbon dioxide were determined, and oxygen consumption (VO_2_) and carbon dioxide production (VO_2_) were calculated. During all tests expired air measurements were made continuously for 20 min and an average of the final 15 min of data was used for the calculation of non-protein respiratory exchange ratio (RER). The ratio of the volume of carbon dioxide produced to the volume of oxygen consumed was used to compute the RER, VO_2_ and VCO_2_ for each minute. The deviation around the mean was allowed to be no more than 10% for VO_2_ and VCO_2_ alongside an RER <0.7 or >1 to be confirmed as steady-state [38]. Daily resting metabolic rate was calculated from oxygen consumption and carbon dioxide production by the Weir formula [41].

#### 2.6.2. Body Composition

##### Air Displacement Plethysmography (BOD POD)

Body composition was assessed by air displacement plethysmography, using the BOD POD (BOD POD ^®^, Life Measurement Instruments, Concord, CA, USA). After voiding participants were asked to change into tight-fitting swimwear which was the same during each visit. All jewellery was removed and participants were provided with a tightly fitted cap to eliminate the presence of trapped isothermal air [42]. A two-point calibration was performed as described previously by McCrory et al. [43]. Body volume was determined by the mean volume of air displaced when the participant was secured in the chamber. Thoracic gas volume (V_TG_) was predicted using the age-specific equation developed by Crapo et al. [44]; this was then used to adjust measured body volume estimated according to the methods described by the manufacturer [45]. The corrected body volume was used in combination with body mass to determine body density, from fat, lean weight and lung volume using the Siri Equation [46].

#### 2.6.3. Meal Tolerance Test

##### Standardised Meals Familiarisation

Following body composition analysis, participants consumed the standardised breakfast meal provided at the onset of the meal tolerance test. Familiarisation was completed to confirm participants were happy to consume this meal during the experimental trials. Alongside the standardised breakfast meal familiarisation, participants were provided with 15 g of protein mixed with 400 mL of water to confirm they would be able to consume this amount and type of nutritional supplement if allocated to RT + PRO. Participants were also familiarised to the pasta meal to be provided at ad-libitum on completion of the meal tolerance test (see Figure 3).

### 2.7. Breakfast Test Meal

The standardised breakfast meal was calculated based on the total daily energy requirements of each participant. Daily energy requirement of the participants was calculated as resting metabolic rate multiplied by a physical activity level factor. A physical activity level factor of 1.40 was applied, representing the activity level of an individual who is predominantly chair bound during the experimental trial [47]. Twenty-two percent of total daily energy requirements were provided within this meal with a macronutrient distribution of 59% CHO, 27% fat and 14% protein (mean = 447 ± 88 kcal) to replicate reported EI and macronutrient distribution in older adults at breakfast meals [48]. The breakfast meal was jam on toast and consisted of white bread (Hovis, thick sliced), unsalted butter (Sainsbury’s), strawberry jam (Sainsbury’s) and semi-skimmed milk.

### 2.8. Subjective Appetite Sensations

Hunger, satiety, fullness and prospective food consumption were assessed by 100 mm visual analogue scales (VAS) [49]. The baseline measure was obtained immediately prior to the standardised breakfast meal (t = 0). After the completion of the standardised breakfast meal, participants completed another VAS, with subsequent measures obtained every half an hour for 180 min (see Figure 3). A composite subjective appetite score was calculated using the following formula: composite appetite score = (hunger + prospective food consumption + (100 − fullness) + (100 − satisfaction)/4) [50]. This single composite score was used for ease of data analysis and presentation, as it has been shown that, with the original six question VAS technique [51], the scores for each question co-vary to a large extent [50]. A higher value is associated with a greater appetite sensation. All visual analogue scales were measured in duplicate by the same researcher to ensure accuracy.

### 2.9. Blood Sampling

A 22-gauge cannula was inserted into the antecubital vein on arrival to the laboratory. After the insertion of the cannula, the participant remained relaxed for 10 min before the fasting sample was taken. This procedure has been shown to stimulate the vagus nerve, which can affect appetite related-hormones leading to reduced appetite sensations [52]. Samples were obtained prior to the ingestion of the standardised breakfast meal (fasted) and then every 30 min following the initiation of the standardised breakfast meal for 180 min. Cannulas were kept patent by flushing with 5 mL non-heparinised saline (0.9% sodium chloride, Baxter Healthcare Ltd., Norfolk, UK) after each sample and at regular intervals between sampling.

To avoid dilution of blood samples with saline, the first 2 mL of each sample drawn was discarded. Blood samples were drawn in 10 mL volumes into pre-chilled tubes containing the anticoagulant K_2_EDTA (Vacutainer, Becton Dickinson, NJ, USA). Tubes were inverted 15 times to ensure mixing with anticoagulant. To prevent any extraneous influences from postural changes, all blood samples were collected whilst the participant was seated [53].

Immediately after collection, samples were centrifuged (ALC, PK130R, Winchester, VA, USA) for 10 min at 1500× *g* and a temperature of 4 °C. The supernatant of each sample was then removed, separated into 1 mL cryovials and initially stored at −40 °C. Samples were then transferred to −80 °C for later analysis.

### 2.10. Assessment of Energy Intake

#### 2.10.1. Ad Libitum Energy Intake

After the completion of the MTT and cannula removal, EI was assessed using a standardised *ad* libitum pasta meal. The *ad libitum* meal was designed to closely align with the UK dietary guidelines [54] for macronutrient proportions (52% carbohydrate, 34% fat and 14% protein). The pasta-based meal consisted of penne pasta (Sainsbury’s), cheddar cheese (Sainsbury’s), tomato sauce (Sainsbury’s) and olive oil (Sainsbury’s). Pasta was cooked for 15 min in unsalted water at 700 W before being mixed with the remaining ingredients and re-heated for 2 min at 700 W. Participants consumed the lunch in isolation to avoid any social influence on food intake. No other visual or food cues were present. A bowl of the aforementioned meal was provided by an investigator and participants were instructed to eat until ‘comfortably full’, with no time limit set for eating. This bowl was replaced before the participant had emptied it to prevent the completion of a bowl signalling satiety, with minimal interaction and this process continued until the participant was comfortably full. Food intake was calculated as the weighted difference in food before and after eating [55]. Water was available at *ad libitum* during participants’ first trial and standardized for each subsequent trial.

#### 2.10.2. Daily Energy Intake

Daily EI was measured at baseline and at Week 12 using three-day weighed food diaries. Participants were provided with instructions on how to complete the diary which included the type, quantity, the preparation of cooking and timing of this intake. The three-day food diary involved the analysis of three consecutive days including at least one weekend day. Upon receiving a completed food diary from a participant, a member of the research team initially checked the clarity and detail of information provided. The participant was then asked to provide any additional clarity or detail required for dietary analysis. The food diary was analysed using online software analysis (Nutritics Ltd., Dublin, Ireland).

### 2.11. Maximal Strength

The multiple maximum strength test method has been demonstrated to be suitable for prescribing the intensity of strength training [56]. The reproducibility of 5-RM strength testing in older adults has been validated [57], and therefore the use of 5-RM testing was employed in this thesis. The order in which the resistance exercises were conducted matched the order during the resistance training intervention and familiarisation. The 5-RM protocol was adapted from Haff and Triplett [58]. The test was initiated with 10 repetitions. This was reduced with each set and a load of 5–10% was added. Once participants had reached 5 repetitions the load was continuously added at the end of each set. Based on the participants’ RPE scores during the familiarisation trial, a starting weight which was estimated to score 5 on the RPE scale was determined by the researcher. Participants were allocated at least four minutes’ rest period before the next attempt to allow sufficient recovery [59]. A 5-RM attempt was confirmed when participants provided an RPE of 10 referring to “very, very heavy- maximum exertion”. If the participant could not complete five repetitions with good technique, then the previous weight was used for the 5-RM score.

One-repetition maximum value for the resistance exercise was then predicted using the formula by Mayhew and Ball [35]:$$1-RM=\;\frac{W}{(52.2+41.9e^{-0.55\cdot R)}/100}$$

This formula was chosen since it was evidenced to have high relative accuracy and low absolute error providing a safe range 1-RM value for older adults starting a resistance training programme [60].

### 2.12. Analysis of Blood Samples

The enzyme-linked immunosorbent assay (ELISA) technique was used to determine leptin and insulin concentrations (ELISA kit, Millipore, MA, USA). To eliminate interassay variation, all samples from each participant were analysed on the same plate when using the plate reader. The sensitivity of these ELISA kits were 0.2 ng/mL and 1 μU/mL respectively and the within batch coefficients of variation were; leptin 4.1% and insulin 4.9%.

### 2.13. Statistical Analysis

With expected adherence rates of 69%, a sample of 46 participants (23 per arm) allowed for the completion of a minimum of 30 participants (*n* = 15 per arm). The study finished with 39 participants (CON = 20 and RT + PRO = 19) All descriptive data were analysed using SPSS for Windows version 26.0 software (SPSS, Chicago, IL, USA). Baseline differences between groups were assessed using Welch’s t-test for all outcome measures. To compare differences between the experimental group (RT + PRO) and control (CON), a linear mixed-effects model analysis was used [61]. Group (CON or RT + PRO), trial (baseline, week 6 and week 12) and time (0 to 120 min) were used as repeated categorical variable for measures that were repeated within a trial (VAS and blood analytes) [62,63,64].

Post-hoc analysis was completed on significant fixed effects using the Bonferroni adjustment and supplemented with Cohen’s *d* effect sizes interpreted as ≤0.2 trivial, >0.2 small, >0.6 moderate, >1.2 large, >2 very large and >4 extremely large [65]. Descriptive data analysis, individual change scores and 95% confidence intervals were calculated for baseline and week 12. Appetite values were compared with the minimal clinically important difference [49]. All blood parameters were compared for fasting concentrations and postprandial area-under-the-curve (AUC). AUC was calculated using the trapezoidal method.

Following the results from the linear mixed models, analyses were undertaken to examine the extent to which the effects of group on changes from baseline to Week 12 in EI were accounted for by changes from baseline to Week 12 in FFM, FM, RMR, and postprandial AUC subjective appetite. *Ad libitum* EI was used as the dependent variable due to the known limitations in underreporting using free-living measures such as food diaries [66]. To assist in identifying variables that may account for the effects of group effects on changes in *ad libitum* EI, we applied a procedure equivalent to the four assumptions of mediation analysis explained previously by [65,66]. Using ANOVA and ANCOVA produces identical results to regression analyses and are conceptually equivalent to regression [67]. First, the independent variable (i.e., group) should affect the dependent variable. Second, the independent variable should affect the mediating variable(s). Therefore, MANOVAs with follow-up ANOVAs were conducted to initially determine group differences for changes in EI (dependent variable) as well as FFM, FM, RMR, and postprandial AUC subjective appetite. The last two criteria are that the mediating variable(s) should affect the dependent variable when adjusting (or controlling) for the independent variable, and the effect of the independent variable on the dependent variable should be reduced in the presence of the mediating variable(s). To address the final two criteria ANCOVAs were conducted. Specifically, we examined the change in the main effect of the independent variable (group) on the dependent variable (change in *ad libitum* EI) after adjusting for differences in the covariates (i.e., changes in FFM, FM, RMR, and AUC postprandial subjective appetite). The magnitude of change in effect (calculated as η^2^) associated with the group factor when a variable is added as a covariate reflects the extent to which this variable (covariate) accounts for group differences in EI [68]. Therefore, we used the above assumptions of mediation to help provide a conservative approach in determining which variables would be worth including as covariates in these analyses, and thereby consider as having a noteworthy contribution in accounting for the effects of the intervention on changes in *ad libitum* EI. Based on evidence that males and females exhibit similar appetite, EI, and gut hormone responses to exercise- and diet-induced energy deficits [64], data from both sexes were combined for analysis.

Based on evidence that males and females exhibit similar appetite, EI, and gut hormone responses to exercise- and diet-induced energy deficits [69], data from both sexes were combined for analysis.

## 3. Results

After initial contact from 184 respondents, further explanation of the experimental trials and screening took place. Participants were excluded for either non-medical based reasons, which included: a history of or current smoker, actively attempting/attempted to lose weight within the last 6 months, could not commit to the full 15-week trial, or taken part in resistance exercise within the last 6 months. Participants were excluded on a medical basis if they were taking medication or had any conditions known to affect appetite regulation. The first 46 participants, who were eligible to participate and verbally agreed to take part in the study, were selected for participation in the 15-week study. Thirty-nine participants completed the full 15 weeks and all experimental trials (see Figure 1).

### 3.1. Compliance

Participants, who completed the intervention exercised at least 92% (22 out of 24) of the expected sessions (average 94 ± 4%). Participants also consumed at least 93% (157 out of 168) protein portions (average 96 ± 2%). Reasons for missed trials included holidays (maximum missed sessions = 2). Protein was missed due to forgetting to take provided portions.

### 3.2. Baseline Paramenters

The only between group difference that was identified at baseline was fasted insulin concentrations. Fasted insulin was significantly lower in RT + PRO compared to CON at baseline (see Table 2). There were also no differences in maximal strength between groups across all exercises (≥0.186).

#### 3.2.1. Fat-Free Mass (FFM)

A significant group*trial interaction (*p* = 0.002) and trial effect was observed for FFM (*p* = 0.004). However, there were no group effects (*p =* 0.582) Fat-free mass significantly increased in RT + PRO from baseline to Week 12 (+1.2 kg; *p* = 0.046; *d* = 0.13; 95% CI = −1.899 to −0.592), and from Week 6 to Week 12 (+0.7 kg; *p* < 0.001; *d* = 0.07; 95% CI = −1.317 to −0.592). There was no change in FFM from baseline to Week 6 in RT + PRO (*p* = 0.097). There were no changes in FFM in CON from baseline to week 6 (*p* = 1.000) or week 12 (*p* = 0.284).

#### 3.2.2. Fat Mass (FM)

There was no significant effect for group (*p* = 0.202), trial (*p* = 0.226) and no significant group*trial interaction effect (*p* = 0.168, see Figure 4) for fat mass.

#### 3.2.3. Body Weight

There was a significant group*trial interaction effect (*p* = 0.021) and trend for trial main effect (*p* = 0.060). There was no group effect (*p* = 0.936). An increase from baseline to Week 12 was observed in RT + PRO (+0.5 kg, *p* = 0.092; *d* = −0.04; 95% CI = −1.108 to 0.059). Weight significantly increased in RT + PRO from baseline to week 6 (+0.8 kg; *p =* 0.002; *d =* −0.30; 95% CI = 0.262 to 1.429). However, there was no difference in weight from week 6 to week 12 of the intervention in RT + PRO (−0.3 kg, *p* = 0.545; *d* = 0.00; 95% CI = −0.262 to 0.904). There were no differences in CON from baseline to Week 6 (*p* = 1.000) and baseline to Week 12 (*p* = 1.000; Figure 5).

#### 3.2.4. Resting Metabolic Rate

There was no significant main effect for group (*p =* 0.143), trial (*p* = 0.668) and no significant group*trial interaction effect (*p* = 0.283; Figure 6) for daily resting metabolic rate.

### 3.3. Subjective Appetite

Fasted subjective appetite did not reveal main effects for group (*p* = 0.768), or a group*trial interaction effect (*p* = 0.629). However, there was a significant effect across trial (*p* = 0.002). Subjective appetite sensations decreased from baseline to Week 12 (−6 mm·min^−1^; *p* = 0.007; *d* = −0.50; 95% CI = −13 to −2 mm·min^−1^) and Week 6 to Week 12 (−5 mm; *p* = 0.008; *d* = 0.43; 95%CI = −13 to −2 mm) irrespective of group (See Table 3). However, no changes occurred from baseline to week 6 (*p* = 1.000). Postprandial appetite perceptions for AUC revealed a significant group*trial interaction (*p* = 0.047) and significant group effect (*p* = 0.027). Postprandial subjective appetite was significantly higher in RT + PRO compared with the CON at Week 12 (+8 mm·min^−1^; *p* = 0.004; *d* = 0.40; 95% CI = 5 to 23 mm·min^−1^). In the RT + PRO postprandial subjective appetite increased significantly from baseline to Week 12 (+8 mm; *p* = 0.012; *d* = 0.24; 95% CI = −13 to −1 mm·min^−1^) and Week 6 to Week 12 (+5 mm·min^−1^; *p* = 0.042; *d* = 0.19; 95% CI = −9 to −2 mm·min^−1^). No differences were observed from baseline to week 6 (*p* = 0.350). Postprandial subjective appetite did not differ across baseline, week 6 and week 12 in CON (*p* = 1.00 for all; see Table 4).

### 3.4. Appetite-Relate Hormone Concentrations

There were no main effects revealed for group (*p* ≥ 0.198), trial (*p ≥* 0.244), or group*trial interaction (*p ≥* 0.100) for fasted (Table 3) or postprandial (Table 4) leptin and insulin circulating concentrations.

### 3.5. Energy Intake (EI)

#### 3.5.1. *Ad Libitum* EI

A group*trial interaction (*p* = 0.040) and trial effect were observed for *ad libitum* EI (*p* = 0.012). There was no main effect for group (*p* = 0.218). The RT + PRO showed a significant increase in *ad libitum* EI from baseline to week 12 (+119 kcal; *p =* 0.001, *d* = 0.60; 95% CI = −41 to 197 kcal) and week 6 to week 12 (+80 kcal; *p* = 0.042; *d =* 0.36; 95% CI = 2 to 158 kcal). There was no change in *ad libitum* EI from baseline to week 6 (*p* = 0.673). There was no difference in *ad libitum* EI across trials in CON from baseline to Week 12 (*p =* 0.803) or Week 6 to Week 12 (*p =* 1.000; see Figure 7).

#### 3.5.2. Daily EI

There was a significant group*trial interaction for daily EI (*p* = 0.010). However, there was no group (*p =* 0.689) or trial (*p* = 0.505) effects. Daily EI increased from baseline to Week 12 in RT + PRO (mean difference = +133 kcal; *d* = 0.29; 95% CI = 1729 to 2245 kcal; Figure 8; Table 5). The CON decreased from baseline to Week 12 (mean difference = −223 kcal; *d* = 0.41; 95% CI = 1753 to 2269 kcal).

#### 3.5.3. Explaining Group Difference in *Ad Libitum* EI

Prior to mediation analyses, a MANOVA was conducted to test for differences in change scores from baseline to week 12 (∆; calculated as Week 12—baseline values) in *ad libitum* EI, postprandial subjective appetite, FM, FFM and RMR. This analysis revealed a multivariate effect of group, *F*(5, 33) = 3.907, *p* = 0.007, η^2^ = 0.372. Separate follow-up ANOVAs identified a significant group effect for ∆ ad labium EI (*F*(1) = 12.789, *p* = 0.001, η^2^ = 0.257), ∆ FFM (*F*(1) = 10.568, *p* = 0.002, η^2^ = 0.222) and ∆ postprandial subjective appetite (*F*(1) = 5.393, *p* = 0.026, η^2^ = 0.127), but not for ∆ FM or ∆ RMR (*p* > 0.05). This supports the outcome of the linear mixed model which demonstrated that FFM and postprandial subjective appetite (AUC) were significantly higher in RT + PRO. Therefore, only ∆ FFM and ∆ postprandial subjective appetite were included as potential mediators in subsequent analysis.

ANCOVAs were conducted with group as the independent variable, ∆ *ad libitum* EI as the dependent variable, and with ∆ FFM and ∆ postprandial subjective appetite as covariates in separate ANCOVAs. When adjusting for ∆ FFM, the effect of group on ∆ *ad libitum* EI remained significant (*F*(1) = 6.052, *p* = 0.034), but the effect was reduced (∆ η^2^ = 0.138). As such, when adjusting for ∆ FFM, the amount of variance in ∆ *ad libitum*, EI attributed to the group (i.e., treatment) dropped from 25.7% to 11.9%, indicating that 54% of the observed treatment effect on ∆ *ad libitum* EI can be explained by ∆ FFM (see Table 6). However, ∆ postprandial subjective appetite did not satisfy the criteria to be considered as having a noteworthy contribution in accounting for the effect of the intervention on ∆ *ad libitum* EI. This, therefore, suggests that ∆ FFM partially mediated the effect of the intervention on ∆ *ad libitum* EI. The relationship is further evidenced by a significant correlation between ∆ *ad libitum* EI and ∆ FFM (*r* = 0.527, *p* = 0.001). However, a separate ANCOVA revealed ∆ postprandial subjective appetite was not a significant covariate (*p* = 0.508) of the group effect on ∆ *ad libitum* EI. Suggesting that ∆ postprandial subjective appetite (proposed mediator) did not affect ∆ *ad libitum* EI (dependent variable) when adjusting for group (independent variable). Therefore, ∆ postprandial subjective appetite did not satisfy the criteria for mediation.

### 3.6. Maximum Strength

There was a significant group*trial interaction (all *p* < 0.001; See Table 7), group main effect (all *p* ≤ 0.030) and effect for trial for all seven exercises (all *p* <0.001). Strength scores increased from baseline to Week 12 in RT + PRO; however, there was no change in CON.

## 4. Discussion

This study investigated the effects of increasing fat-free mass on resting metabolic rate, subjective appetite, appetite-related hormone concentrations and EI in older adults. The primary findings are that 12-weeks of resistance training and protein supplementation led to a significant increase in FFM, ad-libitum EI, daily EI and postprandial subjective appetite perceptions in older adults. These changes were significant from week 6 to week 12 but not from baseline to week 6 of the intervention. However, these differences cannot be attributed to resting energy expenditure as resting metabolic rate was unaffected. Analysis did confirm that increases in EI and appetite were associated with the increase in FFM, with 54% of the change in *ad libitum* EI attributed to the increase in FFM alone. Fasted subjective appetite sensations, and appetite-related and metabolic hormones in both the fasted and postprandial states remained unchanged following the 12-week intervention. Maximum strength was also significantly improved in the RT + PRO.

The majority of studies conducted so far have explored the efficacy of exercise interventions from an obesity perspective, whereby reduction in weight and fat-mass in sedentary individuals are primary objectives [21]. Exercise intervention studies are either associated with negligible [70,71,72] or reduced [73,74] energy consumption after intervention completion despite significant reductions or a maintained fat-mass. Nevertheless, cross-sectional studies have demonstrated associations between FFM and EI [11,17,75]. Despite these inconsistent findings, the results of the present study observed a longitudinal association between changes in FFM and *ad libitum* EI. To our knowledge, this is the first study which supports the speculation that an increase in FFM alters the sensitivity to the appetite control system by augmenting the drive to eat. We can safely also speculate that an increase in FFM with weight gain may play a more important role in exerting feedback signals, which drive EI over fat mass since in our study only a modest and non-significant reduction in fat mass was observed on completion of the intervention. Increased *ad libitum* and daily EI were amplified concurrently to the significant increases in FFM (+1.2 kg) providing further support to this speculation. Nevertheless, the increase in daily EI should be interpreted with a degree of caution since the increase in habitual EI appears to be primarily due to the additional energy received through the supplement. Daily EI increased by an average 46 kcal per day, thus the increased calorific intake did not surpass the calorific content of the whey protein supplement provided (+169 kcal per day). Protein has been shown to exert satiating effects and can often reduce overall EI [76] and is typically used as a weight loss strategy for overweight individuals. Whilst the current study indicates that participant’s dietary EI irrespective of the protein supplements was reduced, overall EI was still heightened as a result of the intervention.

Our findings suggest that the tonic drive to eat is not underlined by resting energy expenditure given that resting metabolic rate remained unaltered following an increase in FFM. This is somewhat surprising considering that Campbell et al. [77] demonstrated a significant increase in resting metabolic rate following 12 weeks of resistance training alongside protein supplementation. Some differences in methodology may account for these discrepancies. For example, the participants attended three sessions per week creating a greater exercise stimulus, which results in a 1.4 kg gain in FFM. In addition, their participants were also untrained upon recruitment to the study. The assessment of resting metabolic rate was also undertaken in the postprandial state; therefore, the thermic effect of food would have been a confounding factor in their study. There is also some evidence to suggest that energy expenditure drives FFM-induced EI through specific lean tissue masses/organs, in particular, the skeletal muscle [78]. This hypothesis was formed based on the ‘aminostatic’ theory [79,80] of appetite control and a ‘protein-static’ [81] control of food intake in which the lean tissue would drive the metabolic requirement to maintain its mass. Whilst appetite and EI did increase concurrently to the increase in FFM, the present study cannot attribute this to resting metabolic rate, which remained unchanged. Weise et al. [75] reported that FFM itself was associated with several brain regions involved in homeostatic appetite control following the tomographic measurement of regional cerebral blood flow in healthy adults. Therefore, it is speculated that the increase in appetite and EI in the present study could be a reflect centrally mediated mechanisms rather than resting energy expenditure per se.

Exercise training studies investigating appetite-related outcomes are crucial to determine the effectiveness of exercise and body composition for weight control. Whilst some research has explored the effect of 12-weeks aerobic exercise on appetite-related variables [26,70,71,72,73,82,83], to date, only one study has explored the effects of 12 weeks supervised resistance training on appetite-related outcomes [84]. Resistance training showed no differences in hunger, fullness, active ghrelin, PP or PYY, on the other hand, leptin had significantly reduced. However, the extent to which this occurs in older adults from both an exercise and body composition perspective is yet to be unveiled. This is of particular importance due to the reductions in appetite, EI and muscle mass associated with ageing [4,6]. The results of the present study strongly suggest that *ad libitum* EI was increased as a result of the increases in FFM. This is supported by the non-significant increase in both *ad libitum* EI and FFM from baseline and week 6 of the intervention. This conflicts with Guelfi, Donges and Duffield [84] who demonstrated no differences in perceived hunger or fullness in the postprandial state following 12 weeks of resistance exercise. However, this study was conducted in previously sedentary overweight and obese middle-aged men and it is suggested that the coupling between EI and energy expenditure in this population is impaired [85]. The participants in the present study were at the lower end of overweight (BMI = 25.7 ± 3.6 kg·m^−2^) and also previously active prior to the intervention, therefore the coupling between EI and energy expenditure are unlikely to be impaired. Nonetheless the data presented from the present study identify a role for FFM in appetite-related outcomes.

In addition, it has been proposed that fat mass and FFM may exert ‘passive’ and ‘active’ signals which drive appetite under different energy balance states [18]. The results from the present study indicate that the RT + PRO were in a positive energy balance given this is a requirement for increases in body weight and skeletal muscle mass [86]. However, it is proposed that during an energy deficit, FFM becomes an active driver of the tonic drive to eat to protect losses in lean tissues [78]. Research suggests that it is not perhaps resting metabolic rate itself that creates a tonic pull on EI at or close to EB, but the potential energy deficit that it can produce. It is apparent that even in a positive energy balance that FFM may act as an active signal to increase EI in older adults; however, this could not be attributed to changes in resting metabolic rate. The large gains seen in 5-RM strength concurrent to the significant increases in FFM indicate that the changes of FFM were likely driven be increases in skeletal muscle tissue. Therefore, increases in *ad libitum* EI may be driven actively by the skeletal muscle mass directly. More research is required about how EI changes under conditions of energy surfeit and deficit to confirm this.

The dose of exercise itself did not appear to substantially influence acute appetite responses. Exercise can raise resting values of energy expenditure several fold depending on the volume, mode and intensity [5]. It has been concluded that a single bout of exercise does affect the immediate post-exercise appetite response [23] including resistance exercise in older adults (unpublished data). However, after 9–16 days of repeated exercise research suggests a partial compensatory response begins to appear [24,25]. The present study observed no differences in subjective appetite sensations, EI and resting metabolic rate after 6 weeks of training whereby significant increases in FFM had not yet occurred. It is speculated that the energy expenditure from resistance exercise may not be sufficient in stimulating appetite compensatory responses.

The hormones measured in the present study were based on the results of our recent meta-analysis [6], which indicated that leptin, insulin, PYY and CCK were elevated in older adults compared to younger adults. The data of the present study demonstrated that the intervention did not affect circulating concentrations of both fasted and postprandial leptin and insulin compared to their baseline values. Many studies have focused on leptin and insulin in response to exercise training. The consensus of these studies suggest that leptin is reduced after both aerobic exercise, even when there were no changes in fat mass observed [87] nor resistance exercise where fat mass was significantly reduced [84], conflicting the findings of the current study. Whereas the results of insulin vary [70,87]. Discrepancies between the present study and Guelfi et al. [84] are likely due to the populations investigated. Many training studies typically recruit inactive individuals at baseline such as Guelfi and colleagues [84]; however, the present study was observed in older adults who were already active. The modulating effect of habitual activity on appetite-related hormones after an exercise training intervention is unknown. However, the differences observed in aerobic training studies are likely to originate from the differences in energy expenditure. A small collection of studies have shown increased acylated ghrelin, PYY, GLP-1 and PP following aerobic interventions [26,70,87]; however, similarly to current study the evidence after resistance training on appetite-related hormones excluding leptin are negligible [84]. Given that CCK and PYY is secreted in response to protein and also, exists in greater circulating concentrations in older adults when compared to young, this warrants future research.

Despite the novel findings in the present study, some notable limitations must be acknowledged. Firstly, in the present study participants were supervised for every resistance training session by a qualified trainer; however, this is unlikely in real-life scenarios due to cost implications, location and motivation. Compliance would likely pose an issue in a real-world setting if a resistance training programme was conducted individually. Additionally, resistance exercise requires good knowledge of anatomy and technique, which not all individuals possess, to minimise injury risk Future research should explore the effects of free-living or home-based interventions on muscle mass and appetite-related variables in older adults to determine effectiveness as well as efficacy. This study was also conducted in active and healthy older adults, who may not represent a true reflection of the older population on the whole. The effects may be different in older adults, who hold underlying comorbidities or diseases, since this may further implicate appetite regulation. Finally, the present study failed to assess total daily energy expenditure and habitual physical activity, due to limitations in free-living measures [88]; therefore, we can only speculate that exercise was not sufficient in stimulating compensatory changes in EI due to little differences in energy expenditure.

## 5. Conclusions

In conclusion, 12 weeks of resistance training and protein supplementation led to a significant increase in FFM, postprandial subjective appetite and *ad libitum* EI in older adults. There was a significant increase in *ad libitum* EI, which appeared to be mainly driven by the increase in FFM. In addition, the mechanisms which underpin increased postprandial subjective appetite and EI in older adults are unknown given that resting metabolic rate and appetite-related hormones concentrations remained unaltered. Based on the findings of the present study chronic resistance training and increased protein intake pose a successful strategy for enhancing FFM and EI, and breaking the vicious cycle between, appetite, EI, and muscle mass reductions in older adults.

## Figures and Tables

**Figure 1 nutrients-13-00141-f001:**
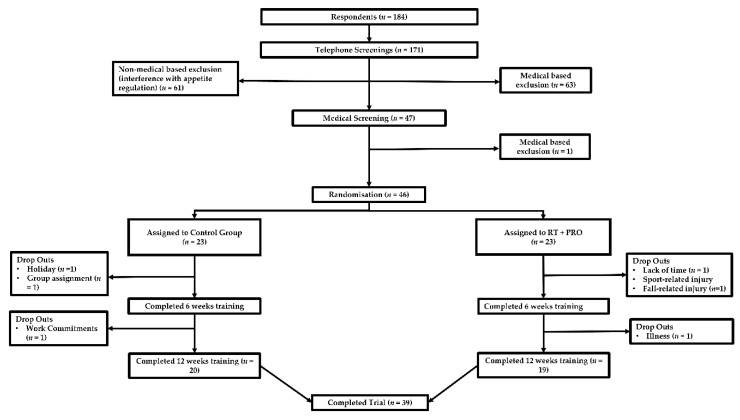
Participant flow from initial respondents to study completion. * Drop-outs related to training and/or testing. Injury and illness were not as a result of the intervention.

**Figure 2 nutrients-13-00141-f002:**
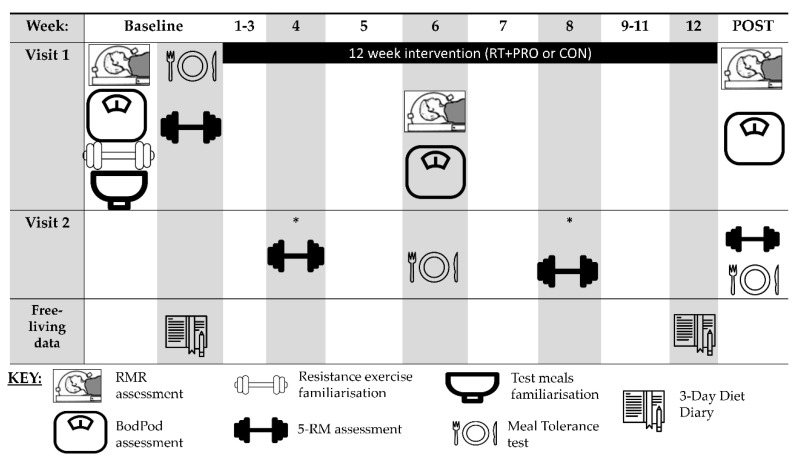
Schematic of the full randomised control trial including intervention and testing days. Participants were assigned to either the RT + PRO or CON during weeks 1–12. * indicates measures taken in RT + PRO only to adjust for training load.

**Figure 3 nutrients-13-00141-f003:**
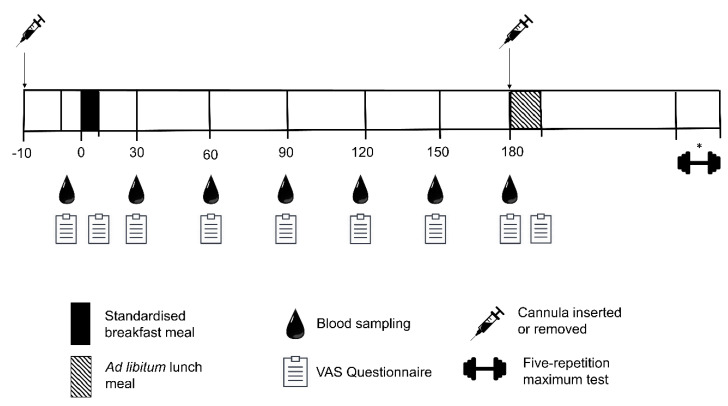
Schematic representation of meal tolerance test conducted at baseline, week 6 and 12. * Indicates procedures that were taken during baseline and Week 12 only. VAS; visual analogue scale.

**Figure 4 nutrients-13-00141-f004:**
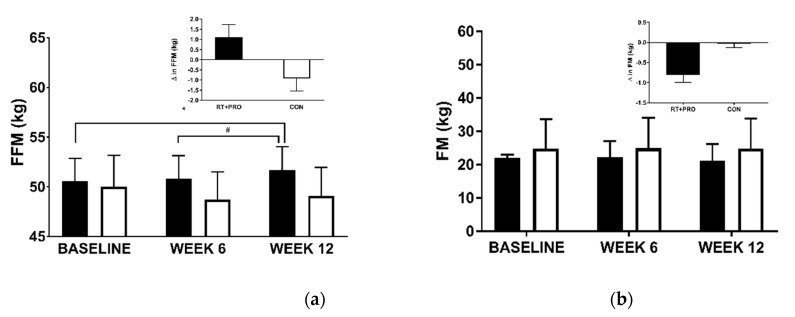
FFM (**a**) and FM (**b**) in CON (white bars; *n* = 20) and RT + PRO (black bars; *n* = 19) at baseline week, 6 and 12. An Asterisk (*) represents a significant difference from baseline to Week 12, a hashtag (#) represents differences from Week 6 to Week 12. The insert graph shows the change from baseline to Week 12. Values are presented as mean ± SE. FFM = Fat-free mass.

**Figure 5 nutrients-13-00141-f005:**
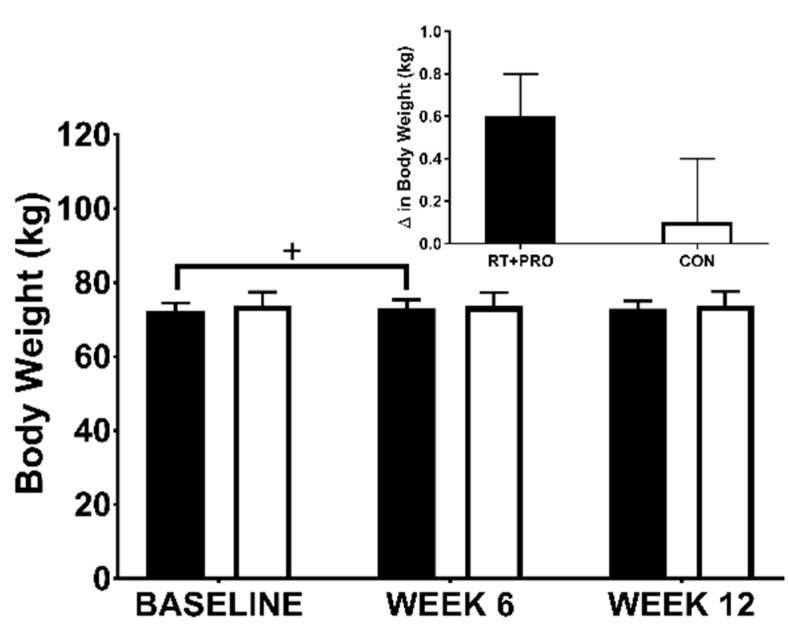
Body weight in CON (white bars; *n* = 20) and RT + PRO (black bars; *n* = 19) groups during baseline week 6 and 12. A addition symbol (+) represents differences from baseline to Week 6. The insert graph shows the change from baseline to Week 12. Values are presented as mean ± SE.

**Figure 6 nutrients-13-00141-f006:**
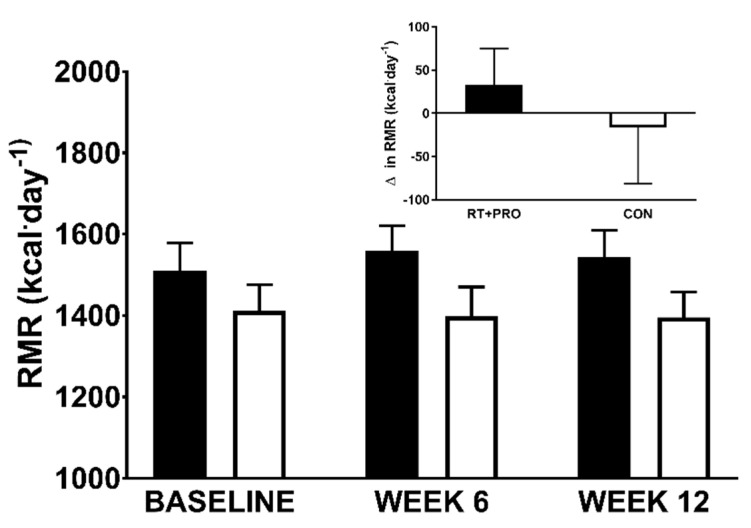
RMR in CON (white bars; *n* = 20) and RT + PRO (black bars; *n* = 19) groups during baseline week 6 and 12. + represents differences from baseline to Week 6. The insert graph shows the change from baseline to Week 12. Values are presented as mean ± SE. RMR = resting metabolic rate.

**Figure 7 nutrients-13-00141-f007:**
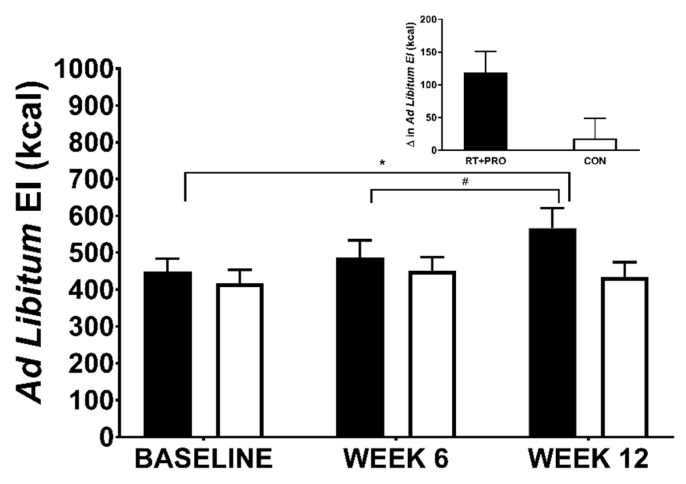
*Ad libitum* EI in CON (white bars; *n* = 20) and RT + PRO (black bars; *n* = 19) groups during baseline, week 6 and 12. An Asterisk (*) represents a significant difference from baseline to Week 12, a hashtag (#) represents differences from Week 6 to Week 12. The insert graph shows the change in *ad libitum* EI from baseline to Week 12. Values are presented as mean ± SE. EI = Energy Intake, RT + PRO = resistance training and protein group, CON = control group.

**Figure 8 nutrients-13-00141-f008:**
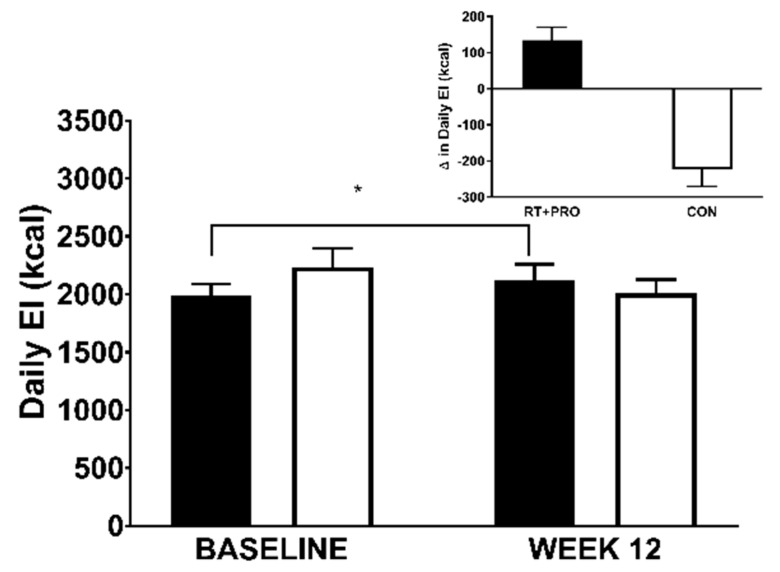
Daily EI in CON (white bars; *n* = 15) and RT + PRO (black bars; *n* = 15) groups for baseline and week 12. An Asterisk (*) represents a significant difference from baseline to Week 12. Values are presented as mean ± SE. EI = Energy Intake, RT + PRO = resistance training and protein group, CON = control group.

**Table 1 nutrients-13-00141-t001:** Participant Characteristics (Mean ± SD).

	RT + PRO (*n* = 19)	CON (*n* = 20)
Age (years)	67 ± 4	65 ± 4
Height (m)	1.67 ± 5.3	1.68 ± 8.4
Body mass (kg)	72.4 ±9.6	73.8 ± 16.5
BMI (kg·m^−2^)	25.7 ± 3.6	26.2 ± 6.1

Abbreviations: SD: Standard Deviation, RT + PRO: Resistance training and protein group, CON; control group, BMI: Body Mass Index.

**Table 2 nutrients-13-00141-t002:** Baseline characteristics of the resistance training and protein supplementation (RT + PRO) and control (CON).

	RT + PRO	CON	*p*-Value
Fat-free mass (kg)	50.5 ± 2.2	49.2 ± 2.8	0.773
Body Weight (kg)	72.4 ± 9.3	73.8 ± 16.1	0.905
Fat mass (kg)	24.7 ± 9.1	21.9 ± 5.3	0.336
Resting metabolic rate (kcal/day)	1543 ± 67	1395 ± 65	0.297
*Ad libitum* energy intake (kcal)	448 ± 42	417 ± 41	0.543
Daily energy intake (kcal/day)	1987 ± 122	2234 ± 128	0.211
Fasted subjective appetite (mm) Subjective appetite AUC (mm·min^−1^)	67 ± 16 44 ± 15	66 ± 15 39 ± 14	0.620 0.205
Fasted leptin (pg·mL) Leptin AUC (pg·mL·min^−1^)	20.2 ± 22.4 16.8 ± 17.7	25.4 ± 30.2 21.4 ± 27.9	0.549 0.551
Fasted insulin (μIU·mL^−1^) Insulin AUC (μIU·mL·min^−1^)	43.0 ± 11.6 13.2 ± 5.2	50.6 ± 17.2 19.4 ± 45.0	0.039 * 0.117

Values are mean (SE), *n* = 20 (CON) and *n* = 19 (RT + PRO). An asterisk (*) indicates significant differences between groups.

**Table 3 nutrients-13-00141-t003:** Fasted subjective appetite sensations and circulating hormones concentrations at baseline, week 6 and Week 12 during the meal tolerance test in RT + PRO and CON.

	Baseline	Week 6	Week 12	Δ from Baseline to Week 12
Subjective appetite (mm)				
RT + PRO	71 ± 16	65 ± 17	66 ± 15	5
CON	75 ± 15	67 ± 18	65 ± 14	10
Leptin (pg·mL^−1^) RT + PRO	20.2 ± 22.4	17.5 ± 17.8	27.6 ± 44.6	7.4
CON	25.4 ± 30.2	27.3 ± 33.9	22.6 ± 27.4	−2.8
Insulin (µIU·mL^−1^) RT + PRO	13.2 ± 5.2	14.7 ± 6.6	16.4 ± 12.4	3.2
CON	19.4 ± 16.6	22.9 ± 22.0	18.2 ± 15.4	−1.2

Values are mean ± SD, CON; *n* = 20 and RT + PRO; *n* = 19.

**Table 4 nutrients-13-00141-t004:** AUC (150 min) for subjective appetite sensations and circulating hormones concentrations at baseline, Week 6 and Week 12 during the meal tolerance test in RT + PRO and CON.

	Baseline	Week 6	Week 12	Δ from Baseline to Week 12
Subjective appetite (mm·min^−1^)				
RT + PRO	44 ± 15	47 ± 12	52 ± 14	8 *
CON	39 ± 14	39 ± 17	44 ± 17	5
Leptin (pg·mL·min^−1^)				
RT + PRO	21.4 ± 27.9	15.6 ± 15.3	19.4 ± 20.4	−2.0
CON	16.8 ± 17.7	14.8 ± 12.3	15.3 ± 15.7	−1.5
Insulin (µIU·mL·min^−1^)				
RT + PRO	43.0 ± 11.6	48.6 ± 17.2	42.9 ± 14.3	−0.1
CON	50.6 ± 17.2	47.6 ± 15.2	45.0 ± 20.6	−4.4

Values are mean ± SD, CON; *n* = 20 and RT + PRO; *n* = 19. An Asterisk (*) represents a significant difference from baseline to Week 12.

**Table 5 nutrients-13-00141-t005:** Daily energy intake and macronutrient intake across groups with and with the protein supplementation for RT + PRO.

	Baseline	Week 12	Δ from Baseline to Week 12
**Energy Intake (kcal)** RT + PRO (total) RT + PRO (-protein supplement) CON	1987 ± 396 1987 ± 396 2234 ± 634	2120 ± 533 1953 ± 507 2010 ± 448	+133 +34 −224
**Protein Intake (g)**			
RT + PRO (total) RT + PRO (-protein supplement) CON **Carbohydrate Intake (g)** RT + PRO (total) RT + PRO (-protein supplement) CON **Fat Intake (g)** RT + PRO (total) RT + PRO (-protein supplement) CON	85 ± 21 85 ± 21 85 ± 18 228 ± 70 228 ± 70 235 ± 54 88 ± 17 88 ± 17 112 ± 34	103 ± 26 73 ± 20 77 ± 19 214 ± 78 210 ± 92 262 ± 65 101 ± 28 97 ± 19 80 ± 17	+18 −12 −8 −14 −18 +27 +13 +9 −32

Values are mean (SE), *n* = 20 (CON) and *n* = 19 (RT + PRO).

**Table 6 nutrients-13-00141-t006:** Effect of covariates on group differences in *ad libitum* energy intake.

Analysis	F	η^2^	Δ η^2^	*p-*Value
One-way ANOVA				
Group	12.79 *	0.257	-	0.01
One-way ANCOVA, group effect adjusting for				
FFM	4.85 *	0.119	0.138	0.034

FFM = Fat-free mass; η^2^ = Eta squared; An Asterisk (*) represents a significant difference in change scores.

**Table 7 nutrients-13-00141-t007:** Maximum Strength scores from 5-RM, change from baseline to Week 12 in RT + PRO and CON.

	Baseline	Week 12	Δ from Baseline to Week 12 (kg)	Effect Size (*d)*
Seated Row RT + PRO CON	37.2 ± 12.3 33.5 ± 14.1	49.8 ± 17.0 33.4 ± 14.1	12.6 −0.1	1.05
Leg Press RT + PRO CON	97.8 ± 26.4 95.1 ± 38.8	151.9 ± 29.2 97.5 ± 30.8	54.1 0.0	0.78
Chest Press RT + PRO CON	33.5 ± 18.1 25.7 ± 17.6	55.6 ± 24.7 28.7 ± 17.9	22.1 0.0	0.98
Leg Curl RT + PRO CON	43.7 ± 16.7 43.8 ± 17.3	70.9 ± 19.7 40.8 ± 17.3	27.2 −3.0	1.30
Lat Pull Down RT + PRO CON	35.2 ± 12.8 35.7 ± 12.2	48.2 ± 15.3 32.7 ± 12.2	13.0 3.0	0.88
Leg Extension RT + PRO CON	45.0 ± 16.8 43.4 ± 18.3	76.7 ± 20.3 47.6 ± 20.8	31.7 4.2	1.16
Shoulder Press RT + PRO CON	21.4 ± 14.6 15.9 ± 10.1	39.5 ± 20.3 18.5 ± 23.6	18.1 2.6	0.80

Values are mean ± SD, CON; *n* = 20 and RT + PRO; *n* = 19.

## Data Availability

The data presented in this study are available on request from the corresponding author.

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
