# Peer review of "An Increase in Fat-Free Mass is Associated with Higher Appetite and Energy Intake in Older Adults: A Randomised Control Trial"

_nutrients, 2021, doi:10.3390/nu13010141_

Round 1
Reviewer 1 Report
- the introduction part is well written describing the previous research in the field and the importance of the present work. Authors are suggested to check the minor typos and commas (for example, line number 43" thus body weight. [3].")
- more recent references relevant to the work may be added in the introduction
- check line number 86-89 (Following withdrawals (see 87
) 39 healthy, independently-living older adults (22F, 17M; Error! Reference source not found.) were included in the study.) - check this in the whole manuscript "Error! Reference source not found"
- Figure 1 is not mentioned in the text of the manuscript.
- check the content of section 2.9.2. Daily energy intake
- check for the use of figure 2 in the whole manuscript, seems repetitions at several places.
- line number 143-151, check the content and format of text
- 'Subjective appetite sensations' is this part of section 2.7. Breakfast test meal?
- check the contents of the 'Subjective appetite sensations' (line 248-249)
Author Response
Thank you for the revised extension on the manuscript. Within the manuscript upon revision the authors noticed a discrepancy in the results. The statistical analysis was run in individually by three authors to confirm the correct statistical analysis. The authors can now conclude that FFM can contribute for ~54% in the observed change in ad libitum EI following the 12-week intervention. This has been amended throughout the manuscript.
Firstly, thank you for taking the time to review our manuscript, it appears that whilst the document was submitted as a word document the conversion to a PDF may have led to an error with some of the hyperlinks when referring to tables and figures in text. To avoid this from happening again we have now removed the hyperlinks associated with tables and figures and used plain text. Hopefully this has amended the issue and there is no “Error! Reference source not found” in the revised manuscript. All errors related to your comments have been addressed and shown in the highlighter function.
Reviewer 1
- the introduction part is well written describing the previous research in the field and the importance of the present work. Authors are suggested to check the minor typos and commas (for example, line number 43" thus body weight. [3].")
Many thanks for your favourable comments. We are pleased you enjoyed reading our manuscript which as you alluded highlights the significance of the current work. Thanks for bringing to our attention these minor typos. We have checked for and corrected minor typos throughout the manuscript. For example on line adding a comma after thus as per your suggestion [Line 43].
- more recent references relevant to the work may be added in the introduction
Thank you for highlight this we have added some more recent work to the introduction including;
- Leij-Halfwerk et al. (2019) - Prevalence of protein-energy malnutrition risk in European older adults in community, residential and hospital settings, according to 22 malnutrition screening tools validated for use in adults ≥65 years: A systematic review and meta-analysis. [Line 42]
- Hopkins et al., (2018)- Potential effects of fat mass and fat-free mass on energy intake in different states of energy balance [Line 59].
- Blundell et al., (2020) - The drive to eat in homo sapiens: Energy expenditure drives energy intake [ Line 59].
- check this in the whole manuscript "Error! Reference source not found"
Thank you and please accept our apologies for these inconsistencies and errors. We are under the impression that these were caused by figures and table hyperlinks used as well as the refence management system. Relying on automated reference management systems (i.e., Endnote) is a huge asset however it does not appear to be error-free. To address these issues we have removed all hyperlinks to tables and figures and have saved the document as plain text and have reviewed all references one by one to ensure no errors or inconsistencies were present.
- Figure 1 is not mentioned in the text of the manuscript.
Thank you for highlighting this it appears that the conversion has removed some of the in text references. Figure 1 has been mentioned on line 87.
- check the content of section 2.9.2. Daily energy intake
We are a little unsure as to the exact nature of this recommendation. We have, however, revisited the paragraph and re-worded it slightly. We hope this clarifies any uncertainty about the process we followed in measuring daily energy intake.
- check for the use of figure 2 in the whole manuscript, seems repetitions at several places.
Reference to Figure 2 has been removed at lines 119 and 282 to reduce repetition, thank you for this suggestion.
- line number 143-151, check the content and format of text
We believe this has been amended in the manuscript
'Subjective appetite sensations' is this part of section 2.7. Breakfast test meal?
Thank you for highlighting this. We have amended this so that Subjective appetite sensation is now under section 2.8 [Line 238].
8. Line number 143-151, check the content and format of text
This has now been amended within the manuscript
9. 'Subjective appetite sensations' is this part of section 2.7. Breakfast test meal?
Thank you for highlighting this, we have amended this in the manuscript [Line 227].
10.check the contents of the 'Subjective appetite sensations' (line 248-249)
We have proofed this for all errors, again apologies for the inconvienece during the review process.
Thank you for your comments.

Reviewer 2 Report
“An increase in fat-free mass is associated with higher appetite and energy intake in older adults: a randomised control trial” by Johnson KO et al. (nutrients-1019930)
Comments:
In this manuscript, the authors tested the suggested correlations between fat-free mass (FFM) and appetite, energy intake, and resting metabolic rate (RMR) experimentally. They manipulated FFM of healthy older adults (total of 39 final participants) by giving 12-week resistance traing + protein supplementation and meausred the outputs. They found that an increase in FFM is associated with higher appetite and energy intake. The trial is well-designed and the topic is important.
Unfortunately, there are minor issues that need to be addressed.
Minor problems:
#1: The adequacy of the study protocol
Figure 1 indicates that the intervention group (RT+PRO) had dropouts due to 1 injury and 1 illness. The author should clearly state if these incidents are due to the intervention or not.
#2: Lack of statistics in the Tables 3,4,5,9
In order to assess if the claims made by author are vaild, it is important if the data in Tables (especially tables 3 & 4) are showing statistically significant differences. However, nothing is mentioned in the table legends. The authors should clarify it.
#3: Not mentioning some of the tables/figures in the result section
Some data-containing figures/tables are not mentioned in the result section. That makes the reviewer difficult to tell the thoughts of authors based on these data. It makes the reviewer hard to judge if the authors are making proper claim based on their data.
#4: Typos
There are 3 parts that appear to be typos:
Line 87: (see )
Line 382: (see )
Line 537~538: Despite significant reductions in fat mass and maintenance.
#5 Too many reference errors
References are missed in the following lines: 88, 108, 148, 231, 249, 290, 427, 441, 449, 452, 453, 466, 478, 479, 507, 510, 516.
#6 Formatting errors
There are formatting errors in lines 144~148, line 290, and Table5.
I believe that the study was done properly, but it is not presented in the adequate form in the submitted version. The authors should address the above-mentioned minor points.
Author Response
In this manuscript, the authors tested the suggested correlations between fat-free mass (FFM) and appetite, energy intake, and resting metabolic rate (RMR) experimentally. They manipulated FFM of healthy older adults (total of 39 final participants) by giving 12-week resistance traing + protein supplementation and meausred the outputs. They found that an increase in FFM is associated with higher appetite and energy intake. The trial is well-designed and the topic is important.
Unfortunately, there are minor issues that need to be addressed.
Thank you for this favourable review, we are glad you enjoyed reading the manuscript and have address all your minor issues. It appears that whilst the document was submitted as a word document the conversion to a PDF may have led to an error with some of the hyperlinks when referring to tables and figures in text. To avoid this from happening again we have now removed the hyperlinks associated with tables and figures and used plain text. Hopefully this has amended the issue and there is no “Error! Reference source not found” in the revised manuscript. All errors related to your comments have been addressed and shown in the highlighter function. Within the manuscript upon revision the authors noticed a discrepancy in the results. The statistical analysis was run in duplicate by three authors. The authors can now conclude that FFM can contribute for ~54% in the observed change in ad libitum EI following the 12-week intervention. This has been amended throughout the manuscript.
#1: The adequacy of the study protocol
Figure 1 indicates that the intervention group (RT+PRO) had dropouts due to 1 injury and 1 illness. The author should clearly state if these incidents are due to the intervention or not.
Thank you for raising this point. The 2 dropouts in the intervention were not related to the intervention. The authors have amended this in text to reflect this [Line 99].
#2: Lack of statistics in the Tables 3,4,5,9
In order to assess if the claims made by author are vaild, it is important if the data in Tables (especially tables 3 & 4) are showing statistically significant differences. However, nothing is mentioned in the table legends. The authors should clarify it.
Thank you for this comment. We agree that the tables could be clearer regarding statistically significant differences. To aid clarity and interpretation, symbols denoting statistical significance have been added. We hope this now nicely compliments the more detailed statistical information provided in the text. Table 4 [Line 450], Table 6 [Line 499] and Table 7 [Line 512].
#3: Not mentioning some of the tables/figures in the result section
Some data-containing figures/tables are not mentioned in the result section. That makes the reviewer difficult to tell the thoughts of authors based on these data. It makes the reviewer hard to judge if the authors are making proper claim based on their data.
Thank you for this suggestion. We have added intext reference to the relevant figures: Figure 5 [Line 411] and Table 6 [Line 485]
#4: Typos
There are 3 parts that appear to be typos:
Line 87: (see )
Line 382: (see )
Line 537~538: Despite significant reductions in fat mass and maintenance.
Thank you for highlighting these typos. Lines 87 and 382 should now read “(see Figure 1)”. Line 525-527 has been revised for greater clarity.
#5 Too many reference errors
References are missed in the following lines: 88, 108, 148, 231, 249, 290, 427, 441, 449, 452, 453, 466, 478, 479, 507, 510, 516.
Thank you for highlighting this. We have amended this throughout the manuscript.
#6 Formatting errors
There are formatting errors in lines 144~148, line 290, and Table5.
Please refer to above comment.
I believe that the study was done properly, but it is not presented in the adequate form in the submitted version. The authors should address the above-mentioned minor points.
Thank you for this favourable review despite the issues in formatting. We have ensured these have been eradicated to help with clarity.
